# Corrosion Protection of 6061 Aluminum Alloys by Sol-Gel Coating Modified with ZnLaAl-LDHs

Youbin Wang [1,2,*], Qiuyu Huang [1,2], Bingtao Zhou [1,2], Zengyin Yuan [1,2], Yuezhou Wei [1,2] and Toyohisa Fujita [1,2]

1    Guangxi Key Laboratory of Processing for Non-Ferrous Metal and Featured Materials, Guangxi University, Nanning 530004, China; qiuyuhuang0728@163.com (Q.H.); 202010103819@mail.scut.edu.cn (B.Z.); yuanzengyin@126.com (Z.Y.); yzwei@gxu.edu.cn (Y.W.); fujitatoyohisa@gxu.edu.cn (T.F.)
2    School of Resources, Environment and Materials, Guangxi University, Nanning 530004, China
*    Correspondence: wangyoubin@gxu.edu.cn; Tel.: +86 07713392507

**Abstract:** In this work, ZnLaAl layered double hydroxides (LDHs) were prepared by the co-precipitation method, and the ZnLaAl-LDHs nanosheets were embedded in sol-gel coating for the corrosion protection of 6061 aluminum alloys. The structure, morphology, and long-term anti-corrosion performance of sol-gel coating modified with ZnLaAl-LDHs were investigated. The structure and morphology analysis showed that nanosheets of ZnLaAl-LDHs are finer than those of ZnAl-LDHs, with the results suggesting that the La can refine the size of LDHs' nanosheets and improve their nucleation rate. The results of long-term corrosion tests showed that the sol-gel coating with ZnLaAl-LDHs exhibits higher corrosion resistance and better stability compared with the sol-gel coating with ZnAl-LDHs, which indicates that the addition of La enhances the anti-corrosion performance of the LDHs and improves the stability of sol-gel coating with LDHs. Finally, the formation mechanism of ZnLaAl-LDHs and the corrosion mechanism of sol-gel coating with ZnLaAl-LDHs on 6061 aluminum alloys are both discussed in detail.

**Keywords:** ZnAl-LDHs; lanthanum; aluminum alloys; corrosion behavior



## 1. Introduction

Aluminum alloys have found extensive applications in the aerospace industry, power industry, automobile manufacturing, construction, food industry and marine shipbuilding industry due to their high strength to weight ratio, high thermal conductivity and processing, and low cost [1,2]. Unfortunately, when aluminum alloys are exposed to chloride-containing environment, pitting corrosion easily appears [3], as well as intergranular corrosion [4] and stress corrosion cracking [5]. In this case, numerous surface treatments have been studied to protect aluminum alloys, such as anodic oxidation [6,7], polymer coating [8,9], silane coating [10–12] and chemical conversion film technology [13,14]. Among the above technologies of surface treatments, chemical conversion films have attracted considerable attention due to their advantages of low cost and simple operation. As a new chemical conversion film, layered double hydroxides (LDHs) film has the characteristics of environmental friendliness and excellent corrosion resistance. Many scholars have begun to study its application in the corrosion protection of aluminum alloys and find LDHs film can provide excellent protection of aluminum alloys.

Layered double hydroxides (LDHs), also known as like-hydrotalcite, is an inorganic functional material with a structure composed of positively charged octahedrons [15]. Its general formula is $[M(II)_{1-x}M(III)_x(OH)_2]^{x+}(A^{n-})_{x/n} \cdot mH_2O$ [16]. LDHs exhibits similar structure to that of brucite when some of the divalent metal cations M(II) are replaced by the trivalent metal cations M(III). Excess positive charge is compensated by interlaminar anions ($A^{n-} = CO_3^{2-}$, $NO_3^-$, $Cl^-$, etc.). Many methods have been developed for preparing LDHs

film [17], such as the in-situ growth method [18,19], co-precipitation method [20,21], electrodeposition method, etc. [22,23]. Among the above preparation methods, co-precipitation has the advantage of efficiency and low-cost. J.M. Vega et al. [24]. synthesized ZnAl-LDHs film on the aluminum alloys using the co-precipitation method which inhibited the corrosion of the alloys. Zhang Fen [21] et al. also prepared MgAl-LDHs film on AZ31 alloys by co-precipitation method, and demonstrated that the resulting films exhibit good corrosion resistance.

Recently, in order to improve the corrosion protection performance of ZnAl-LDHs film, some scholars attempted to insert rare earth elements into ZnAl-LDHs. The Ce-doped ZnCeAl-LDHs was prepared and it was found that the ZnCeAl-LDHs has better corrosion protection performance [25–27] than that of ZnAl-LDHs. In previous works, our research group [28] prepared ZnLaAl-LDHs by the in-situ growth method, and found that the La can refine ZnLaAl-LDHs nanosheets and improve the corrosion protection performance. LDHs can also be used as corrosion inhibitors to enhance the active anticorrosion of the sol-gel coatings [29,30]. According to the authors, the corrosion protection performance of sol-gel coating modified with ZnLaAl-LDHs has not been systematically studied.

In the present work, the sol-gel coatings with La-doped ZnAl-LDHs were prepared on the surface of 6061 aluminum alloys. Then, titrator, scanning electron microscopy (SEM), X-ray diffraction (XRD) and X-ray photoelectron spectroscopy (XPS) were used to study the effect of doped La content on the microstructure of ZnAl-LDHs prepared by co-precipitation. In order to understand the anticorrosion behavior of sol-gel coatings with ZnLaAl-LDHs, the 6061 aluminum alloys with LDHs/sol-gel coatings were evaluated by Electrochemical impedance spectroscopy (EIS) in 3.5 wt.% NaCl solution. Finally, the corrosion protection mechanism of the ZnLaAl-LDHs/sol-gel coating on the 6061 aluminum alloys was discussed in detail.

## 2. Experimental

### 2.1. Materials and Preparation

A total of 6061 aluminum alloys were cut into specimens with the size of $\Phi$ 14 mm $\times$ 5 mm. The specimens were ground step by step with 200#, 600#, 800#, 1200# SiC papers until the surface had no obvious scratches, and then polished with 0.1 $\mu$m diamond slurry. Finally, ultrasonic cleaning was conducted with high pure water and blow-drying.

Configuration of solution: Zn, Al nitrates, and Zn, Al, La nitrates were dissolved in 100 mL (the total concentration is 0.75 M, which M (II)/M (III) = 2:1), respectively. The solution of Zn, Al, and La nitrates had the ration of $La^{3+}/Al^{3+} = x$ ($x$ = 1/5, 1/3 and 1/1). Then, the above solution was added to 0.1 M $NaNO_3$ solution (200 mL, pH = 10.0) drop by drop under vigorous stirring. The pH of the reaction process was controlled to 10.5 $\pm$ 0.2 by continuous addition of 2 M NaOH. Nitrogen atmosphere was maintained during the experiment to avoid carbonate anion pollution. The final white colloid was then obtained by placing it at 65 °C for 18 h. Then, the specimens were washed four times with distilled water and filtered. Finally, the washed white colloid was dried at 60 °C for 12 h. The LDHs powders were named ZnAl-LDHs, ZnLaAl-LDHs-1/5, ZnLaAl-LDHs-1/3 and ZnLaAl-LDHs-1/1, respectively.

The mixed organic–inorganic sol-gel coating was configured as the carrier of LDHs, and the sol-gel coating were prepared by two different sols: the first sol was prepared by mixing and stirring propyl trimethyl silicane (GPTMS), ethanol and deionized water at a volume ratio of 1:3.5:1 (240 r/min, 1 h); The second sol was prepared by mixing and stirring ethanol, acetic acid and zirconium n-propoxide (TPOZ, propanol, 70% propanol) at a volume ratio of 3:2:1 (240 r/min, 1 h); the last step involved mixing and stirring the above two sols (240 r/min, 0.5 h). The final sol was divided into two parts: (a) One part of the sol was left without any treatment and coated on 6061 aluminum alloys substrate named sol-gel; (b) the other part of the sol was stirred with 3 mg/mL LDHs powders. The 6061 aluminum alloys substrate coated with LDHs/sol-gel coating named ZnAl-LDHs/sol-gel, ZnLaAl-LDHs-(1/5, 1/3 and 1/1)/sol-gel. To prepare the 6061 aluminum alloys with

sol-gel coating, the alloys substrate soaked in the final sol-gel solution for 5 min, and then drawn out at a speed of 10 cm/min. These specimens were stood in air for 15 min, and then dried at 60 °C for 3 h.

The 6061 aluminum alloys with sol-gel coating were immersed in 3.5 wt.% NaCl solution at 25 °C for 168 h. Finally, specimens were removed from the NaCl solution, rinsed with distilled water, and then dried at room temperature.

### 2.2. Characterization

Titration experiments were conducted with a titrator (Metrohm Tiamo 905, Metrohm, Herisau, Switzerland) using 1 M NaOH solution as a titration solution, and the drop acceleration was 0.1 mL/min. The pH evolution was recorded using Tiamo 2.5 titration software (Metrohm, Herisau, Switzerland). The test solution was mixed with metallic nitrate solution, with an initial volume of 100 mL.

The LDHs powders were characterized by XRD (D/MAX 2500 V, Rigaku, Tokyo, Japan) using the Cu K$\alpha$ radiation. The scanning rate was 8° min$^{-1}$. The evaluation of the data was done using the Jade 6.5 software (MDI, Burbank, CA, USA). The chemical compositions of LDHs powders were probed using XPS (ESCALAB 250Xi, ThermoFisher, Waltham, MA, USA) with Al K$\alpha$ radiation (1486.6 eV). The experimental data were assessed by XPSpeak4.1 software to analyze the chemical composition of LDHs powders. The morphology and cross section of LDHs powders and LDHs/sol-gel coatings (before and after 168 h immersion) were investigated by SEM (SU8220, Hitachi, Japan).

Electrochemical experiments were carried out using electrochemical workstation (PAR-STAT 4000 A, Ametek, Bowen, PA, USA) in 3.5 wt.% NaCl solution at room temperature. A classic three-electrode system was adopted: the specimens, Pt sheet and a saturated calomel electrode (SCE) were used as the working electrode, counter electrode and reference electrode, respectively. All the specimens were sealed by epoxy resin but leaving the testing surface (1 cm$^2$) uncovered for the corrosion tests. Electrochemical impedance spectroscopy (EIS) was tested with the amplitude of 10 mV and the frequency range from $10^5$ to $10^{-2}$ Hz. Finally, the experimental data were fitted by using ZSimpWin 3.1 software (Ametek, Bowen, PA, USA).

In order to ensure the repeatability of the experiment, all the experiments were conducted three times.

## 3. Results and Discussion

### 3.1. Preliminary Titrations

In order to detect the range of pH that is needed for co-precipitation, the titration experiment of metal salt solution, which was used to prepare LDHs, was conducted. Figure 1 shows the titration curve of different metal salt solutions using 1 M NaOH. It can be seen from the compared titration curves that a plateau appears in all curves at pH values of 3.6 and 5.2, respectively.

The emergence of the plateau at pH = 3.6 is due to Al$^{3+}$ consuming OH$^-$ to produce Al(OH)$_3$ (Equation (1)), and this plateau shortened with the reduction in the Al content (curves of 1 to 4); the emergence of the plateau at pH = 5.2 is due to multiple action (Equations (2)–(5)): Zn reacts with OH$^-$ to form Zn(OH)$^+$, the Al(OH)$_3$ convers to Al(OH)$_4{}^-$, and the LDHs generate during the process. The La$^{3+}$ participates in the formation of ZnLaAl-LDHs simultaneously during the plateau at pH = 5.2. Moreover, with the addition of La$^{3+}$ (curves 2–4), a supplementary plateau at pH around 8 is observed due to the formation of La(OH)$_3$.

$$Al^{3+} + 3OH^- \rightarrow Al(OH)_3 \tag{1}$$

$$Al(OH)_3 + (OH)^- \rightarrow Al(OH)_4 \tag{2}$$

$$Zn^{2+} + OH^- \rightarrow Zn(OH)^+ \tag{3}$$

$$Al(OH)_4^- + 3Zn(OH)^+ + OH^- + NO_3^- + H_2O \rightarrow ZnAl - LDHs \tag{4}$$

$$Al(OH)_4^- + 4Zn(OH)^+ + La^{3+} + 5OH^- + NO_3^- + H_2O \rightarrow ZnLaAl - LDHs \qquad (5)$$

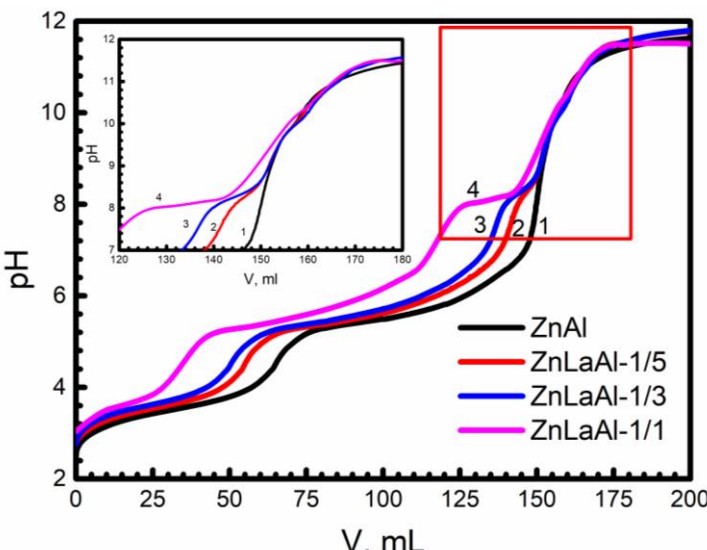

**Figure 1.** Titration curves of the mixed salt solution with adding the titrant (1.0 M NaOH): (1) 0.50 M $Zn^{2+}$ + 0.25 M $Al^{3+}$; (2) 0.50 M $Zn^{2+}$ + 0.25 M ($Al^{3+}$/$La^{3+}$ = 1/5), (3) 0.50 M $Zn^{2+}$ + 0.25 M ($Al^{3+}$/$La^{3+}$ = 1/3), (4) 0.50 M $Zn^{2+}$ + 0.25 M ($Al^{3+}$/$La^{3+}$ = 1/1); the inset is the magnified patterns of the selected area.

### 3.2. Characterization of LDHs Powders

Figure 2 shows the XRD patterns of the different LDHs powders. The (003), (006) and (110) reflection characteristics of the ZnAl-LDHs-NO$_3$ can be observed at 10°, 20° and 60° [31,32], indicating that ZnAl-LDHs-NO$_3$ has been successfully prepared by the co-precipitation method. The impurities of $Zn(OH)_2$ and ZnO are detected in the ZnAl-LDHs powder. At the same time, we found that the $La(OH)_3$ appears after doped La [28], which may be attributed to the fact that not all La cations enter into ZnAl-LDHs. This result is similar to the Ce-doped ZnAl-LDHs [25]. Otherwise, the impurity of $Zn(OH)_2$ in the ZnLaAl-LDHs powder decreases after doped La, and the change of ZnO is not obvious. The above phenomenon illustrates that the addition of La can inhibit the formation of $Zn(OH)_2$.

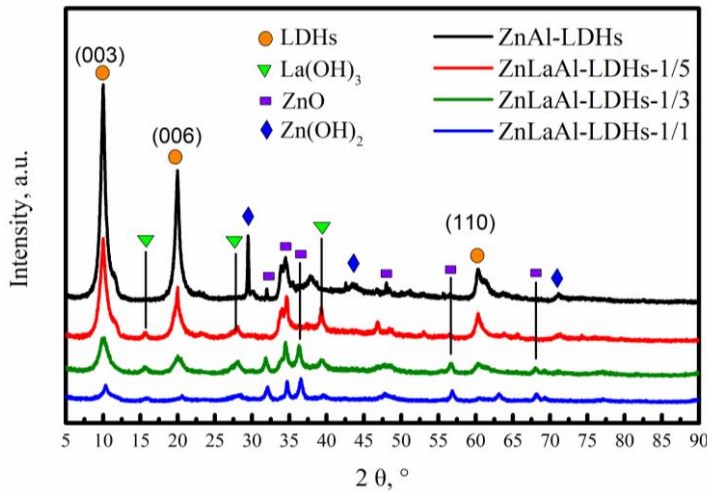

**Figure 2.** XRD spectra of different LDHs powders prepared by co-precipitation method.

XPS has been performed to explore the composition of the different LDHs powders (Figure 3). From Figure 3a, it is obvious that the peak of Zn 2p$_{3/2}$ shifts to the lower binding

energy with the increase in La content. According to the XPS database and literature [33,34], the peaks of ZnO and LDHs occur at 1021.2 and 1022.7 eV, respectively. The shift of the peak of Zn $2p_{3/2}$ indicates that the content of ZnO gradually increases with the increasing of the La content. As can be seen in Figure 3b, the binding energy of Al 2p at 73.8 eV can be attributed to ZnAl-LDHs [35]. In addition, with the increasing of the doped La content during the preparation process, the content of $Al(OH)_3$ in the compound decreases. It can be observed from Figure 3c that the La 3d peak in the spectra of ZnLaAl-LDHs occurs at 835.85 eV, which corresponds to the peak of $La(OH)_3$ [28].

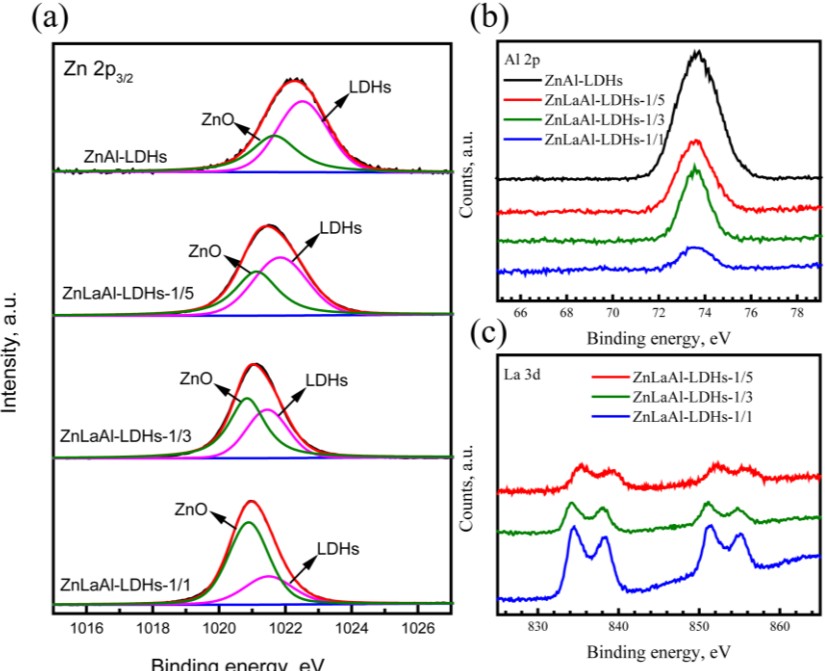

**Figure 3.** The (**a**) Zn $2p_{3/2}$, (**b**) Al 2p and (**c**) La 3d high resolution XPS spectra of the different LDHs powders.

Figure 4 shows the SEM microstructures of different LDHs powders, and the corresponding EDS spectra of ZnAl-LDHs, ZnLaAl-LDHs-1/5, ZnLaAl-LDHs-1/3 and ZnLaAl-LDHs-1/1. The ZnAl-LDHs and ZnLaAl-LDHs powders consists of numerous nanosheets, which agrees with the typical morphology of LDHs prepared by co-precipitation method [26]. Figure $4a_1$–$d_1$ reveal the morphology change of ZnLaAl-LDHs with different La content: the size of the ZnLaAl-LDHs nanosheets becomes smaller after La-doping, and this phenomenon is more obvious with the increasing of the La content. This phenomenon indicates that La plays the role of refining ZnLaAl-LDHs nanosheets and inhibiting the crystal growth of LDHs.

The corresponding EDS spectra of selected areas (A, B, C and D) in different LDHs powders are shown in Figure $4a_2$–$d_2$, respectively. The results show that Zn, Al, N and O elements are detected in ZnAl-LDHs and ZnLaAl-LDHs powder, and La element is just detected in ZnLaAl-LDHs powders, which indicates that the La element successfully inserts in the structure of LDHs during the preparation process of co-precipitation. The [M(III)]/[M(II)] ratio of LDHs is generally between 0.25 and 1.0 [36–38]. The [M(III)]/[M(II)] ratios of ZnAl-LDHs and ZnLaAl-LDHs are both between 0.25 and 1, and the content of La in LDHs was increased with the increasing of the doping content of La.

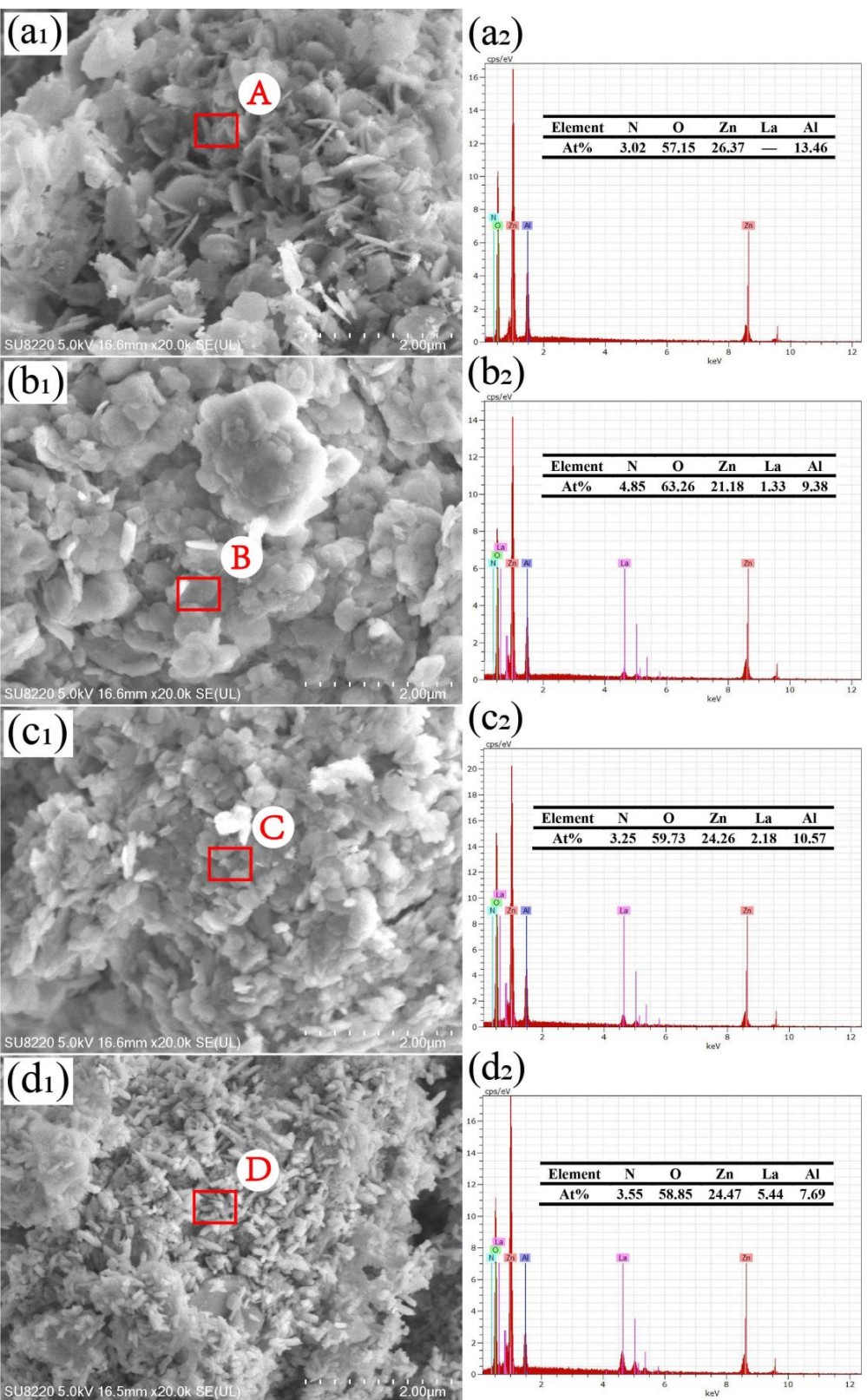

**Figure 4.** SEM morphologies of different LDHs powders prepared by co-precipitation method: (**a₁**) ZnAl-LDHs, (**b₁**) ZnLaAl-LDHs-1/5, (**c₁**) ZnLaAl-LDHs-1/3, (**d₁**) ZnLaAl-LDHs-1/1; and the corresponding EDS spectra of selected areas (A, B, C and D) are shown as (**a₂**–**d₂**), respectively.

### 3.3. Corrosion Behavior

In order to explore the corrosion protection of different LDHs/sol-gel coatings for the 6061 aluminum alloys, Figures 5 and 6 show the surface and cross section morphology of 6061 aluminum alloys with different LDHs/sol-gel coatings immersed in 3.5 wt.% NaCl solution. According to Figure 5a$_1$–a$_5$, the different LDHs powders are evenly distributed in sol-gel coating, and its surface does not have any obvious cracks, pores and other defects. The cross section (Figure 5b$_1$–b$_5$) shows that the thickness of sol-gel coating modified with LDHs powders is about 3.5 μm; there is no obvious differences compared with sol-gel coating modified without LDHs powders (3.2 μm).

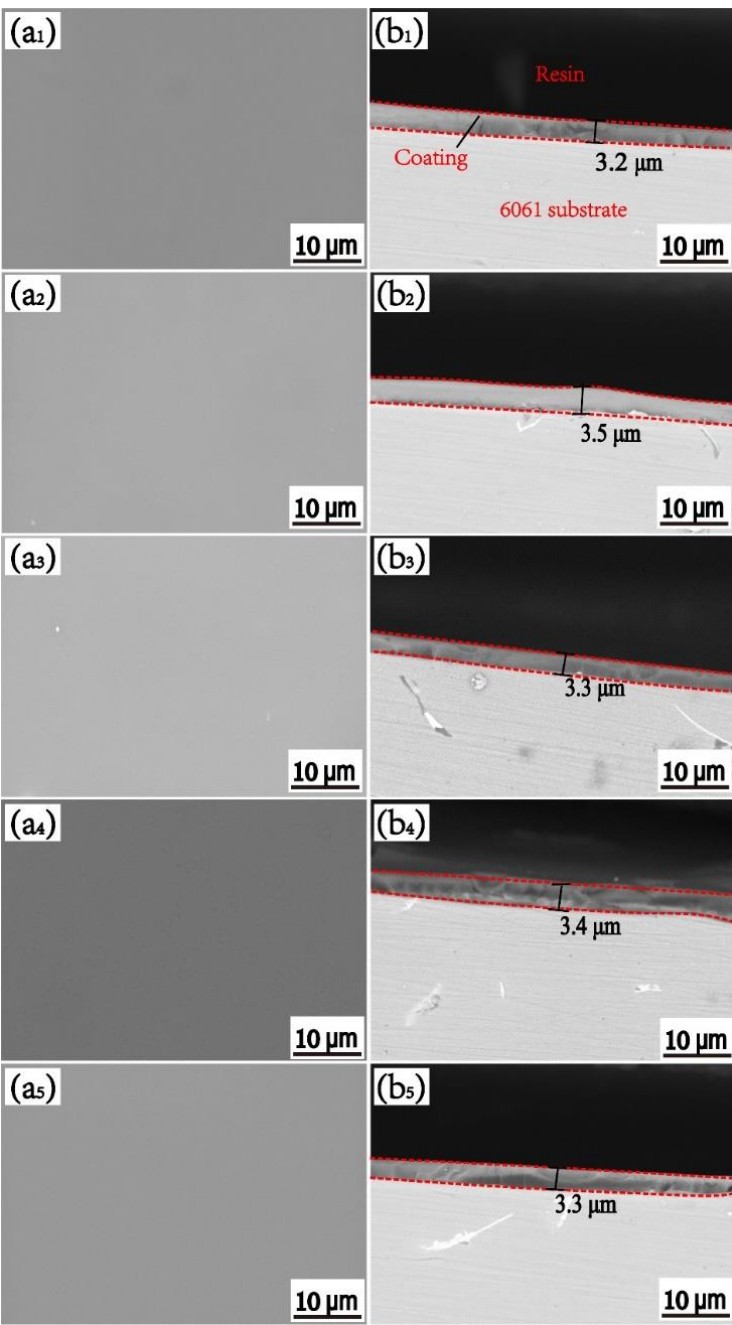

**Figure 5.** The surface and cross-section morphology of 6061 aluminum alloys with different sol-gel coatings immersed in 3.5 wt.% NaCl solution for 0 h, respectively: (**a$_1$**,**b$_1$**) sol-gel coating unmodified, and sol-gel coating modified with: (**a$_2$**,**b$_2$**) ZnAl-LDHs, (**a$_3$**,**b$_3$**) ZnLaAl-LDHs-1/5, (**a$_4$**,**b$_4$**) ZnLaAl-LDHs-1/3, (**a$_5$**,**b$_5$**), ZnLaAl-LDHs-1/1.

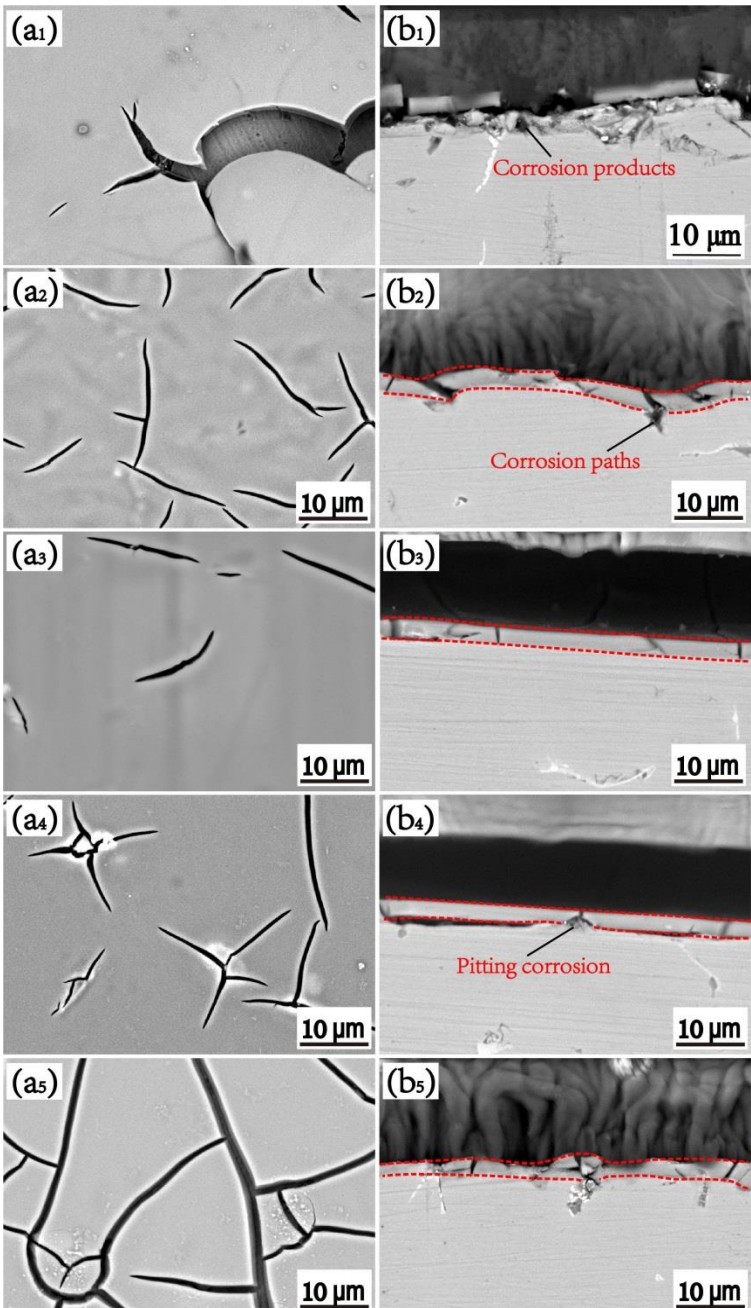

**Figure 6.** The surface and cross-section morphology of 6061 aluminum alloys with different sol-gel coatings immersed in 3.5 wt.% NaCl solution for 168 h, respectively: ($a_1$,$b_1$) sol-gel coating unmodified, and sol-gel coating modified with: ($a_2$,$b_2$) ZnAl-LDHs, ($a_3$,$b_3$) ZnLaAl-LDHs-1/5, ($a_4$,$b_4$) ZnLaAl-LDHs-1/3, ($a_5$,$b_5$), ZnLaAl-LDHs-1/1.

The morphologies of 6061 aluminum alloys with different LDHs/sol-gel coatings after an immersion time of 168 h are shown in Figure 6. The sol-gel coating has an excellent corrosion protection effect for 6061 aluminum alloys. According to Figure 6, corrosion cracks are found in all sol-gel coatings, and the sol-gel coating without LDHs shows exfoliation. It can be also observed from the cross-section morphology of Figure 6$a_1$–$b_1$ that the sol-gel coating without LDHs has obvious fracture phenomenon. The stress corrosion cracking is due to the corrosion products formed on the 6061 aluminum alloys/coating interface during the corrosion process, which pushes the sol-gel coating off the interface and causes the sol-gel coating to lose its protective performance. Compared with sol-gel coating without LDHs, the LDHs/sol-gel (Figure 6) shows better corrosion resistance, the

stress corrosion cracks of the sol-gel coating are greatly reduced, and the corrosion degree of 6061 aluminium substrate is significantly reduced. This phenomenon can be attributed to the addition of ZnAl-LDHs enhancing the stability of the sol-gel coating.

Compared with ZnAl-LDHs/sol-gel, fewer cracks are observed in the ZnLaAl-LDHs-1/5/sol-gel coatings, which is due to the ability of La doped in ZnAl-LDHs to greatly enhance the stability of the LDHs/sol-gel coating. However, the cracks increase gradually with the increasing of the La-doped content. This is due to the excessive addition of La inhibiting the growth of ZnLaAl-LDHs and enhancing the content of ZnO, which results in the uniformity of coating being destroyed. ZnLaAl-LDHs has the dual function of capturing $Cl^-$ anions and releasing corrosion inhibitor, which delays the diffusion of corrosive $Cl^-$ ions to the 6061 aluminum substrate and inhibits the cracking of sol-gel coating.

The EIS of 6061 aluminum alloys with different sol-gel coatings, for different immersion times, is tested to evaluate the long-term corrosion protection performance of the LDHs/sol-gel coatings. Figure 7 shows the equivalent circuits model for EIS data fitting, where $R_{sol}$ is the solution resistance of the electrolyte; $R_{LDHs}$ and $CPE_{LDHs}$ are related to LDHs/sol-gel coating; $R_{OX}$ and $CPE_{OX}$ correspond to the resistance and capacitance of $Al_2O_3$ film, respectively; $R_{ct}$ and $CPE_{dl}$ represent charge transfer resistance and double-layer capacitance, respectively. In order to better fit the experimental data, a $CPE$ is used instead of the capacitor to compensate for the non-homogeneity in the system. The impedance of $CPE$ is calculated by $Z_{CPE} = [Y(j\omega)^n]^{-1}$, where Y is $CPE$ constant, $\omega$ is frequency and $n$ is dimensionless index ($0 \le n \le 1$), respectively. The $CPE$ are pure resistance and pure capacitance, respectively, when values of $n$ are 0 and 1.

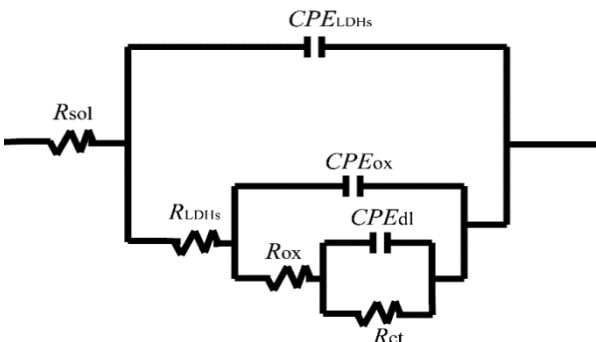

**Figure 7.** Equivalent circuit models for EIS data fitting.

Table 1 shows the EIS results by fitting the equivalent circuit models. The charge transfer impedance ($R_{ct}$) can be used to reflect the reaction activity at the interface of the coating/6061 aluminum substrate. $R_{ct}$ decreases with the increasing of immersion time, indicating that the reaction at the interface of the coating/alloys substrate occurs easily during the immersion process. Specifically, the $R_{ct}$ of LDHs/sol-gel coating is higher than that of sol-gel coating without LDH during the same immersion time, which can be explained by the fact that the interface reactions occur less easily with the addition of LDHs.

The Bode plot diagram (phase angle vs. frequency) is shown in Figure 8, which exhibited three time constants during the immersion process. The first time constant ($10^{-2}$–$10^{-1}$ Hz) is caused by the electrochemical activity during the corrosion process; the second time constant, at the intermediate frequency ($10^0$–$10^2$ Hz), is due to the oxide film between the Al substrate and sol-gel coating; the last time constant (>$10^4$ Hz) corresponds to the electrochemical response of the LDHs/sol-gel coating.

**Table 1.** Fitting parameters of different sol-gel coatings using the equivalent circuit in Figure 7.

| Specimen | Time h | $CPE_{ox}$ $\Omega^{-1}\cdot cm^{-2}\cdot S^{n1}$ Y$_1$ | $n_1$ | $R_{LDHs}$ $\Omega\cdot cm^2$ | $CPE_{ox}$ $\Omega^{-1}\cdot cm^{-2}\cdot S^{n2}$ Y$_2$ | $n_2$ | $R_{ox}$ $\Omega\cdot cm^2$ | $CPE_{dl}$ $\Omega^{-1}\cdot cm^{-2}\cdot S^{n3}$ Y$_3$ | $n_3$ | $R_{ct}$ $\Omega\cdot cm^2$ | $\chi^2$ $10^{-3}$ |
|---|---|---|---|---|---|---|---|---|---|---|---|
| 6061-Blank | 0 | $2.50 \times 10^{-8}$ | 0.92 | 198.60 | $9.92 \times 10^{-7}$ | 0.86 | $5.78 \times 10^2$ | $3.18 \times 10^{-6}$ | 0.91 | $1.63 \times 10^5$ | 2.54 |
|  | 24 | $8.24 \times 10^{-7}$ | 0.93 | 8.76 | $5.57 \times 10^{-6}$ | 0.92 | $3.18 \times 10^4$ | $5.97 \times 10^{-4}$ | 0.74 | $3.51 \times 10^4$ | 2.22 |
|  | 72 | $6.09 \times 10^{-7}$ | 0.86 | 16.14 | $8.23 \times 10^{-6}$ | 0.92 | $2.02 \times 10^4$ | $2.68 \times 10^{-4}$ | 0.62 | $2.96 \times 10^4$ | 1.25 |
|  | 168 | $3.85 \times 10^{-7}$ | 0.91 | 12.13 | $1.34 \times 10^{-5}$ | 0.90 | $1.26 \times 10^4$ | $8.98 \times 10^{-4}$ | 0.80 | $2.39 \times 10^4$ | 0.51 |
| ZnAl-LDHs | 0 | $3.18 \times 10^{-7}$ | 0.82 | 174.60 | $4.31 \times 10^{-6}$ | 0.91 | $2.11 \times 10^5$ | $1.23 \times 10^{-5}$ | 0.92 | $1.11 \times 10^5$ | 1.29 |
|  | 24 | $4.94 \times 10^{-7}$ | 0.89 | 18.12 | $5.49 \times 10^{-6}$ | 0.94 | $7.79 \times 10^4$ | $2.89 \times 10^{-4}$ | 0.85 | $9.52 \times 10^4$ | 1.49 |
|  | 72 | $4.68 \times 10^{-7}$ | 0.92 | 13.51 | $6.18 \times 10^{-6}$ | 0.93 | $4.39 \times 10^4$ | $1.36 \times 10^{-3}$ | 0.75 | $2.95 \times 10^4$ | 1.69 |
|  | 168 | $6.07 \times 10^{-7}$ | 0.91 | 10.23 | $7.84 \times 10^{-6}$ | 0.90 | $1.74 \times 10^4$ | $3.74 \times 10^{-3}$ | 0.87 | $4.54 \times 10^3$ | 1.72 |
| ZnLaAl-LDHs-1/5 | 0 | $3.38 \times 10^{-8}$ | 0.97 | 123.20 | $4.38 \times 10^{-6}$ | 0.93 | $7.05 \times 10^5$ | $8.66 \times 10^{-5}$ | 0.75 | $1.04 \times 10^6$ | 0.60 |
|  | 24 | $3.93 \times 10^{-7}$ | 0.93 | 10.68 | $4.95 \times 10^{-6}$ | 0.94 | $2.87 \times 10^5$ | $4.36 \times 10^{-5}$ | 0.75 | $5.24 \times 10^5$ | 0.78 |
|  | 72 | $4.97 \times 10^{-7}$ | 0.90 | 15.37 | $5.34 \times 10^{-6}$ | 0.93 | $1.57 \times 10^5$ | $6.73 \times 10^{-5}$ | 0.78 | $3.35 \times 10^5$ | 1.24 |
|  | 168 | $6.61 \times 10^{-7}$ | 0.93 | 12.57 | $5.86 \times 10^{-6}$ | 0.93 | $1.05 \times 10^5$ | $9.24 \times 10^{-5}$ | 0.99 | $2.27 \times 10^5$ | 1.30 |
| ZnLaAl-LDHs-1/3 | 0 | $3.24 \times 10^{-8}$ | 1.00 | 48.27 | $5.81 \times 10^{-6}$ | 0.93 | $4.93 \times 10^5$ | $1.25 \times 10^{-4}$ | 0.78 | $1.34 \times 10^5$ | 0.19 |
|  | 24 | $4.52 \times 10^{-7}$ | 0.87 | 19.07 | $6.57 \times 10^{-6}$ | 0.83 | $8.29 \times 10^4$ | $1.50 \times 10^{-4}$ | 0.82 | $1.82 \times 10^5$ | 1.77 |
|  | 72 | $9.69 \times 10^{-7}$ | 0.92 | 3.93 | $8.14 \times 10^{-6}$ | 0.92 | $2.88 \times 10^4$ | $9.99 \times 10^{-5}$ | 0.80 | $8.97 \times 10^4$ | 1.75 |
|  | 168 | $1.31 \times 10^{-5}$ | 0.91 | 4.59 | $1.03 \times 10^{-5}$ | 0.92 | $1.72 \times 10^4$ | $1.57 \times 10^{-4}$ | 0.72 | $6.70 \times 10^4$ | 1.76 |
| ZnLaAl-LDHs-1/1 | 0 | $2.09 \times 10^{-8}$ | 0.98 | 102.90 | $2.92 \times 10^{-6}$ | 0.90 | $3.87 \times 10^4$ | $2.74 \times 10^{-6}$ | 0.93 | $5.07 \times 10^4$ | 0.28 |
|  | 24 | $5.44 \times 10^{-7}$ | 0.88 | 12.72 | $6.04 \times 10^{-6}$ | 0.92 | $5.57 \times 10^4$ | $1.53 \times 10^{-5}$ | 0.77 | $1.02 \times 10^5$ | 1.28 |
|  | 72 | $5.11 \times 10^{-7}$ | 0.91 | 10.87 | $6.80 \times 10^{-6}$ | 0.92 | $2.46 \times 10^4$ | $1.51 \times 10^{-4}$ | 0.76 | $6.64 \times 10^4$ | 1.22 |
|  | 168 | $8.40 \times 10^{-7}$ | 0.91 | 8.49 | $9.13 \times 10^{-6}$ | 0.91 | $1.49 \times 10^4$ | $4.27 \times 10^{-4}$ | 0.97 | $3.15 \times 10^4$ | 1.39 |

Note: Y—capacitance constant; *n*—capacitance exponent.

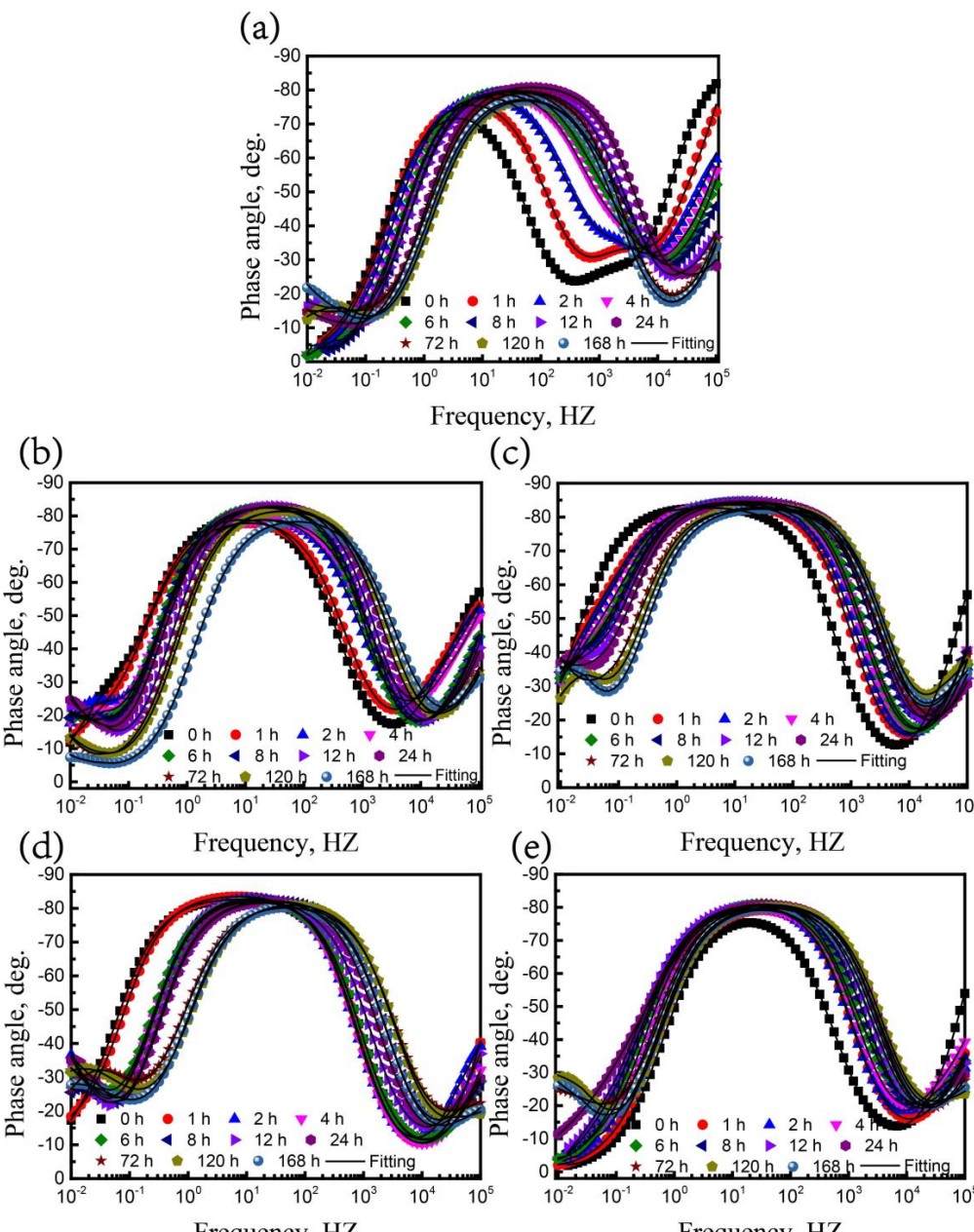

**Figure 8.** Bode plot (phase angle vs. frequency) of 6061 aluminum alloys with blank sol-gel coating (**a**), and sol-gel coating modified with: (**b**) ZnAl-LDHs, (**c**) ZnLaAl-LDHs-1/5, (**d**) ZnLaAl-LDHs-1/3, (**e**) ZnLaAl-LDHs-1/1.

In the test of electrochemical properties, the |Z| at the low frequency of the Bode modulus diagram can be used to estimate the corrosion rate, while the higher |Z| indicates a lower corrosion rate [39]. As can be seen in Figure 9a–c, the |Z| value of sol-gel coating without LDH is lowest in all specimens and the LDHs/sol-gel coatings have higher values of |Z|. Compared with the |Z| value of ZnLaAl-LDHs/sol-gel, it can be found that the corrosion resistance gradually decreases with the increasing of the La content, which is due to the excessive addition of La greatly inhibiting the growth of the ZnLaAl-LDHs nanosheet. Moreover, the content of ZnO increases, which breaks the integrity of the coating and adds coating internal defects.

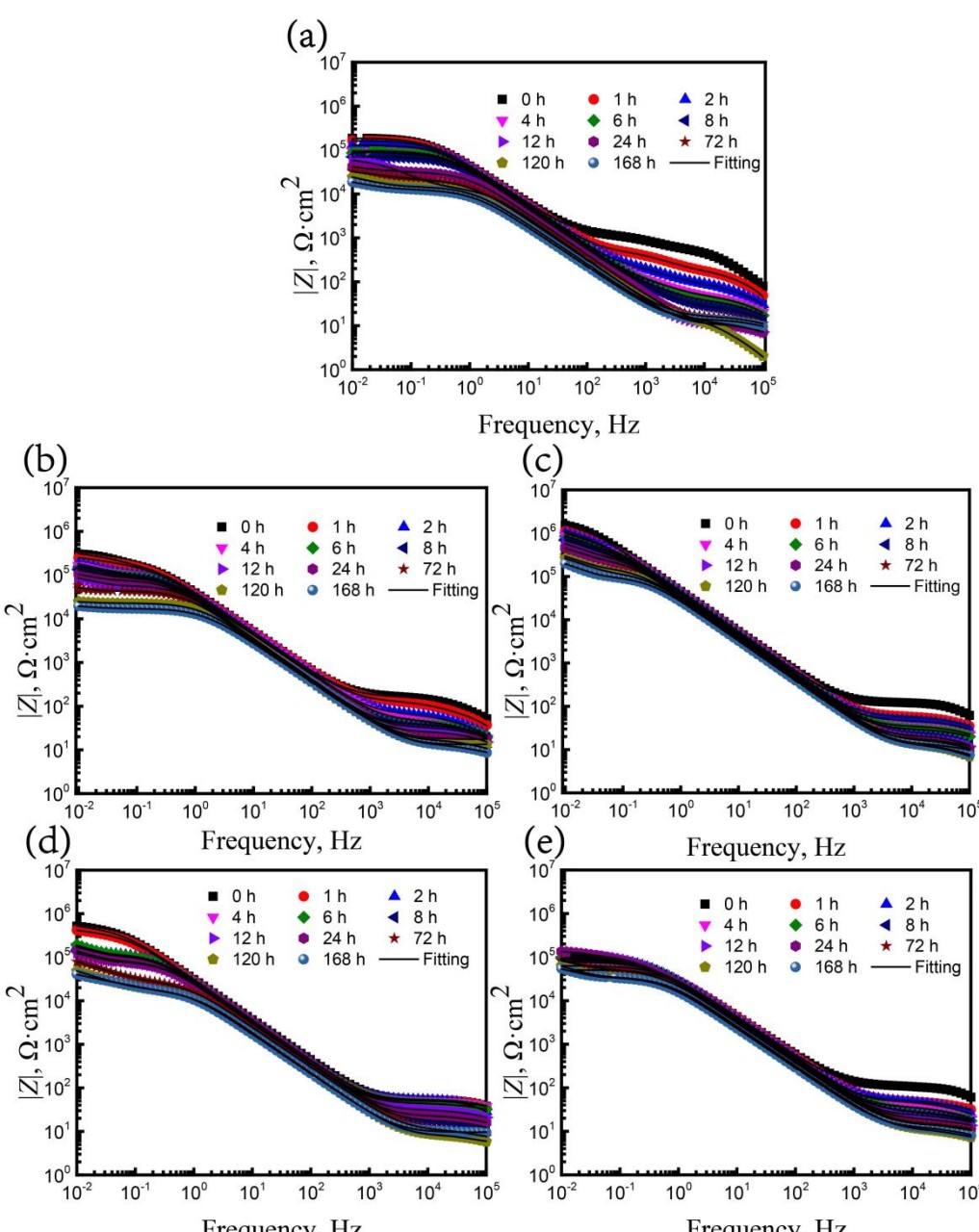

**Figure 9.** Bode plot (modulus vs. frequency) of 6061 aluminum alloys with blank sol-gel coating (**a**), and sol-gel coating modified with: (**b**) ZnAl-LDHs, (**c**) ZnLaAl-LDHs-1/5, (**d**) ZnLaAl-LDHs-1/3, (**e**) ZnLaAl-LDHs-1/1.

The Nyquist diagram (Figure 10) describes the change of impedance of 6061 aluminum alloys with LDHs/sol-gel coating during the corrosion process in 3.5 wt.% NaCl solution. The circle radius of the Nyquist diagram is larger, which illustrates that the corrosion protection of coatings is better [40]. It can be seen from the Nyquist diagram that the corrosion resistance of all sol-gel coatings decreases with the increasing of the immersion time. During immersion times of 0–24 h, the corrosion resistance of ZnLaAl-LDHs-1/1/sol-gel coating increases gradually, which is due to the ZnLaAl-LDHs nanosheets releasing $La^{3+}$ cations to form the $La(OH)_3$, resulting in self-healing behavior. However, the lowest corrosion resistance of ZnLaAl-LDHs-1/1/sol-gel has insufficient protective ability to protect 6061 aluminum substrate. Compared with other specimens, the resistance value of ZnLaAl-LDHs-1/5/sol-gel is highest in all the specimens at the initial stage and after an immersion time of 168 h. The above analysis shows that the La can enhance the

corrosion resistance of ZnLaAl-LDHs/sol-gel coatings and reduce the corrosion rate of the 6061 aluminum substrate, but the excessive addition of La will inhibit the growth of ZnLaAl-LDHs and enhance the content of ZnO, which results in the uniformity of ZnLaAl-LDHs/sol-gel coatings being destroyed and the corrosion resistance decreasing.

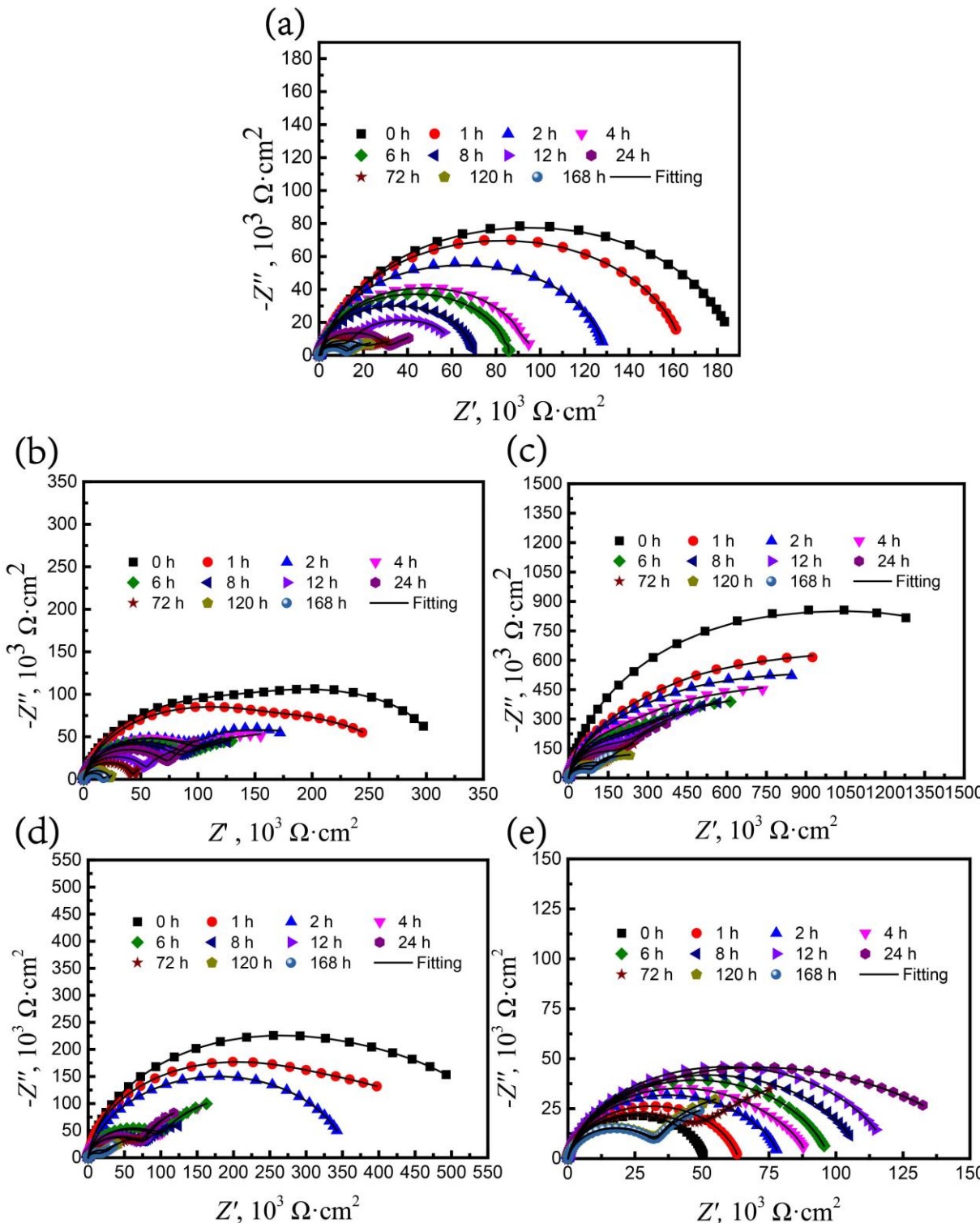

**Figure 10.** Nyquist spectra of 6061 aluminum alloys with different sol-gel coating (**a**) blank sol-gel coating, and LDHs/sol-gel coating: (**b**) ZnAl-LDHs, (**c**) ZnLaAl-LDHs-1/5, (**d**) ZnLaAl-LDHs-1/3, (**e**) ZnLaAl-LDHs-1/1.

### 3.4. Corrosion Mechanism

The corrosion protection mechanism of ZnLaAl-LDHs/sol-gel coating is proposed in Figure 11. Due to its density, the ZnLaAl-LDHs/sol-gel coating can be used as a physical barrier to resist the corrosion of chlorine anions (Figure 11a). In addition, the improvement of corrosion performance of ZnLaAl-LDHs/sol-gel coating is also related to the ion exchange capacity of ZnLaAl-LDHs (Figure 11b), namely the adsorption and retention of chlorine anions and the release of $NO_3^-$ [28]. $Cl^-$ anions are consumed due to the ion exchange property of ZnLaAl-LDHs during the transfer process of $Cl^-$ anions to 6061 aluminum alloys substrate. The ZnLaAl-LDHs/sol-gel coating will show cracks after long-term immersion; the $Cl^-$ anions gradually pass through the gap of the coating, and make contact with the aluminum alloys substrate. The pitting corrosion and corrosion products will appear in the interface between sol-gel coating and substrate, pushing the sol-gel coating off the interface (Figure 11c). The corrosion cell will form $OH^-$ when $Cl^-$ reacts with aluminum alloys substrate, and it combines with $La^{3+}$, which is released from ZnLaAl-LDHs to form the insoluble $La(OH)_3$. Finally, $La(OH)_3$ will form a self-healing film to provide secondary protection for the 6061 aluminum alloys substrate (Figure 11d).

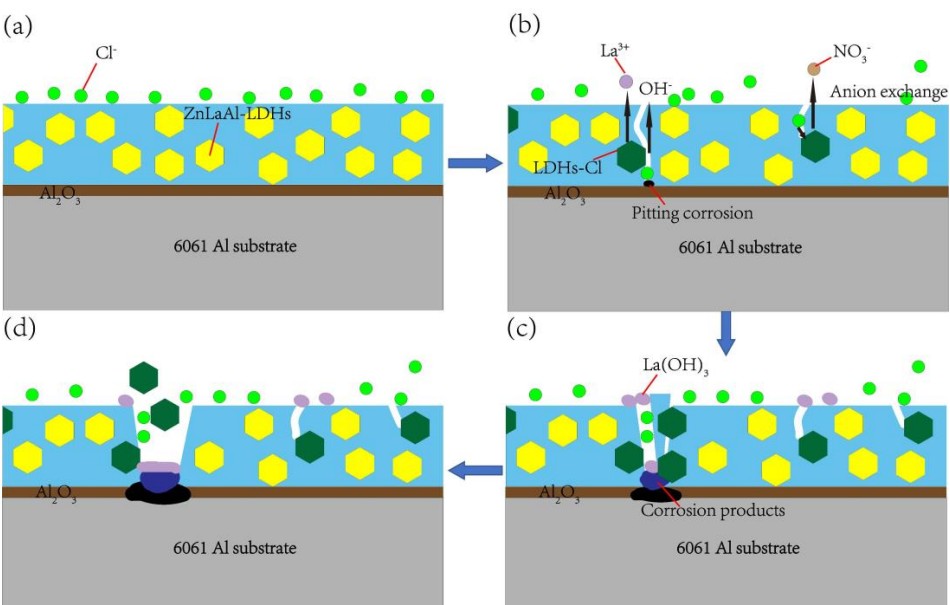

**Figure 11.** The schematic illustration of the corrosion process of ZnLaAl-LDHs/sol-gel coating, (**a–d**) the change of corrosion morphology with the increasing of immersion time.

## 4. Conclusions

The sol-gel coatings with La-doped ZnAl-LDHs were prepared on the surface of 6061 aluminum alloys. The aim of this work was to understand the anticorrosion behavior of sol-gel coatings with ZnLaAl-LDHs on 6061 aluminum alloys. The results show that the size of ZnAl-LDHs nanosheets is obviously reduced after the addition of doped La, and this effect will be more and more obvious with the increasing of the La content, which is due to the fact that La can inhibit the growth of LDHs nanosheets and improve its nucleation rate, resulting in the smaller nanosheets of ZnLaAl-LDHs. The corrosion resistance of sol-gel coating with La-doped ZnAl-LDHs has greatly improved, but the corrosion resistance of ZnLaAl-LDHs/sol-gel coating decreases with the increasing of the La content. This phenomenon is due to the excessive La greatly inhibiting the growth of ZnLaAl-LDHs nanosheets, and $Zn(OH)_2$ and ZnO phases appearing in the precipitate.

The long-term corrosion tests show that ZnAl-LDHs addition can enhance the corrosion resistance of sol-gel coating, and La-doped ZnAl-LDHs can greatly increase this effect. This phenomenon can be explained as follows: (I) the size of ZnLaAl-LDHs nanosheets decreases after La-doping, which causes them to be homogeneously distributed in the

sol-gel coating and, thus, catch more Cl⁻ anions; (II) the La³⁺ cation will be released from the ZnLaAl-LDHs nanosheets during the corrosion process, and it can form the self-healing film of La(OH)₃ to provide secondary protection for 6061 aluminum alloys substrate.

**Author Contributions:** Data curation, Y.W. (Youbin Wang), B.Z. and Z.Y.; formal analysis, Y.W. (Youbin Wang) and B.Z.; funding acquisition, Y.W. (Youbin Wang) and Y.W. (Yuezhou Wei); investigation, Q.H.; methodology, Q.H.; resources, T.F.; supervision, Y.W. (Youbin Wang); writing—original draft, Q.H.; writing—review & editing, Y.W. (Youbin Wang). All authors have read and agreed to the published version of the manuscript.

**Funding:** This work was founded by the Guangxi Natural Science Foundation (Grant No. 2017GXNS FBA198202), the National Natural Science Foundation of China (Grant No. 11975082), and the Guangxi Science and Technology Major Project (Grant No. AA17204100).

**Institutional Review Board Statement:** Not applicable.

**Informed Consent Statement:** Not applicable.

**Data Availability Statement:** The data presented in this study are available in article.

**Conflicts of Interest:** The authors declare no conflict of interest.

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
