# Peer review of "Corrosion Protection of 6061 Aluminum Alloys by Sol-Gel Coating Modified with ZnLaAl-LDHs"

_coatings, doi:10.3390/coatings11040478_

Round 1

Reviewer 1 Report

This paper demonstrated the microstructure and corrosion behavior of ZnAl-LDH and ZnLaAl-LDH on 6061 aluminum alloy.

However, I can find any distinguishable point of this paper comparing with their previous work (ref-23. Effect of lanthanum addition on microstructures and corrosion behavior of ZnAl-LDHs film of 6061 aluminum alloys.

It would be necessary to clearly state what is different from their previous work.

Author Response

     In Ref-23, ZnLaAl-LDH coating was prepared on the surface of 6061aluminum alloy by in-situ growth method. In this work, the ZnLaAl-LDH nanosheets was prepared by co-precipitation method, and then used as corrosion inhibitors to embed in the sol-gel coating on the aluminum alloy. The aim of this work is to understand the anticorrosion behavior of sol–gel coatings with ZnLaAl-LDHs on 6061 aluminum alloys.

      There are obvious differences in the research object and preparation process of the two materials.

Reviewer 2 Report

Review of the “Influence of doped La on the microstructure and corrosion behavior of ZnAl-LDHs sol-gel coating on 6061 aluminum alloys”

The English should be checked by professional service or native English as there are numerous issue starting with the first sentence in abstract

Also check the entire paper for typos

Despite 35 references only 3 are some how recent 2018-2019 which indicate that some major gap in the research literature is and no much novelty !! To adjust this issue I suggest the authors looking into novel references and update your state of art accordingly. It is worth looking into :

https://doi.org/10.1016/j.matdes.2020.109026

In introduction you said sol gel coating and in conclusions “prepared by co-precipitation method” So please be clear

The second paragraph in conclusion is rather fit better to discussion.

Define all the acronyms before their first appearance in text

For how long was used ground for each sand paper??

How many repetition were conducted for each type of trial?/

Line 200 claim that “ has excellent effect of corrosion protection for alloy alloys” it should specified for which type of al alloys !!

In Figure 5 you indicate the same scale for zoomed images so this is incorrect please rectify. Otherwise I suggest to use the other way around in big picture the deposition and then with zoom the details at the coating interface

“diagram (Figure 9) can describe” simple describe and delete “can “

Author Response

Review 2#

  1. Review of the “Influence of doped La on the microstructure and corrosion behavior of ZnAl-LDHs sol-gel coating on 6061 aluminum alloys”. The English should be checked by professional service or native English as there are numerous issue starting with the first sentence in abstract. Also check the entire paper for typos.

Response:

We have worked on the typos and grammatical mistakes of the English, and hope it is now improved a lot. Correction and revision are directly updated in the manuscript.

  1. Despite 35 references only 3 are some how recent 2018-2019 which indicate that some major gap in the research literature is and no much novelty !! To adjust this issue I suggest the authors looking into novel references and update your state of art accordingly. It is worth looking into :

https://doi.org/10.1016/j.matdes.2020.109026

Response:

Thanks for this comment, we have added the following reference:

[9] Pawan Kumar, Brijnandan S. Dehiya, Anil Sindhu, Ravinder Kumar, Catalin I. Pruncu, Anil Yadav. Fabrication and characterization of silver nanorods incorporated calcium silicate scaffold using polymeric sponge replica technique. Mater. Design. 195 (2020): 109026. https://doi.org/10.1016/j.matdes.2020.109026

[17] A. C. Bouali, M. Serdechnova, C. Blawert, J. Tedim, M. G. S. Ferreira, M. L. Zheludkevich, Layered double hydroxides (LDHs) as functional materials for the corrosion protection of aluminum alloys: A review. Appl. Mater. Today. 21 (2020): 100857. https://doi.org/10.1016/j.apmt.2020.100857

[27] Liang Wu, Xingxing Ding, Zhicheng Zheng, Aitao Tang, Gen Zhang, Andrej Atrens, Fusheng Pan. Doublely-doped Mg-Al-Ce-V2O74- LDH composite film on magnesium alloy AZ31 for anticorrosion. J. Mater. Sci. & Technol. 64 (2021): 66-72. https://doi.org/10.1016/j.jmst.2019.09.031

[29] Jianhua Liu, You Zhang, Mei Yu, Songmei Li, Bing Xue, Xiaolin Yin. Influence of embedded ZnAlCe-NO3− layered double hydroxides on the anticorrosion properties of sol–gel coatings for aluminum alloy. Prog. Org. Coat. 81 (2015): 93-100. https://doi.org/10.1016/j.porgcoat.2014.12.015

[30] K. A. Yasakau, A. Kuznetsova, S. Kallip, M.Starykevich, J.Tedim, M. G. S. Ferreira, M. L. Zheludkevich,. A novel bilayer sys-tem comprising LDH conversion layer and sol-gel coating for active corrosion protection of AA2024. Corros. Sci. 143 (2018): 299-313. https://doi.org/10.1016/j.corsci.2018.08.039

  1. In introduction you said sol gel coating and in conclusions “prepared by co-precipitation method” So please be clear

Response:

Thanks for this comment. In this work, the ZnLaAl-LDH nanosheets was prepared by co-precipitation method, and then the ZnLaAl-LDH used as corrosion inhibitors to embed in the sol-gel coating on the 6061aluminum alloy.

In conclusions“The ZnLaAl-LDHs are successfully prepared by co-precipitation method” has been revised as “The sol-gel coatings with La-doped ZnAl-LDHs were prepared on the surface of 6061 aluminum alloys”

  1. The second paragraph in conclusion is rather fit better to discussion.

Response:

We agree with this comment and have corrected the description as the following:

“The long-term corrosion tests show that ZnAl-LDHs addition can enhance the corrosion resistance of sol-gel coating, and La-doped ZnAl-LDHs can greatly increase this effect. This phenomenon can be explained as following: (I) the size of ZnLaAl-LDHs nanosheets decreases after La-doped, which makes it homogeneous distributed in sol-gel coating and catch more Cl- anions; (II) La3+ cation will release from ZnLaAl-LDHs nanosheets during corrosion process, and it can form the self-healing film of La(OH)3 to provide secondary protection for 6061 aluminum alloys substrate.”

  1. Define all the acronyms before their first appearance in text

Response:

We have defined all the acronyms before their first appearance in text

  1. For how long was used ground for each sand paper??

Response:

The polishing time is about 3 minutes for each sand paper, until there are no cross scratches.

  1. How many repetition were conducted for each type of trial?/

Response:

In order to ensure the repeatability of the experiment, all the experiments were conducted for three times. We have added the corresponding description on the experimental section.

  1. Line 200 claim that “ has excellent effect of corrosion protection for alloy alloys” it should specified for which type of al alloys !!

Response:

We have corrected the description as the following:

“The sol-gel coating has excellent effect of corrosion protection for 6061 aluminum alloys.”

  1. In Figure 5 you indicate the same scale for zoomed images so this is incorrect please rectify. Otherwise I suggest to use the other way around in big picture the deposition and then with zoom the details at the coating interface

Response:

We have corrected the Figure 5 and the corresponding description.

Figure 5. The surface and cross-section morphology of 6061 aluminum alloys with different sol-gel coatings: (a1, b1) sol-gel coating unmodified, and sol-gel coating modified with (a2, b2) ZnAl-LDHs, (a3, b3) ZnLaAl-LDHs-1/5, (a4, b4) ZnLaAl-LDHs-1/3, (a5, b5), ZnLaAl-LDHs-1/1.

Figure 6. The surface and cross-section morphology of 6061 aluminum alloys with different sol-gel coatings immersed in 3.5 wt.% NaCl solution for 168 h: (a1, b1) sol-gel coating unmodified, and sol-gel coating modified with (a2, b2) ZnAl-LDHs, (a3, b3) ZnLaAl-LDHs-1/5, (a4, b4) ZnLaAl-LDHs-1/3, (a5, b5), ZnLaAl-LDHs-1/1.

10. “diagram (Figure 9) can describe” simple describe and delete “can “

Response:

We have corrected the description as the following:

“Nyquist diagram (Figure 9) describe the impedance change of 6061 aluminum alloys with LDHs/sol-gel coating during the corrosion process in 3.5 wt% NaCl solution.”

Round 2

Reviewer 1 Report

The distinguishable point between this work and previous work is explained in the revised introduction. 

The alloy was characterized well. 

Reviewer 2 Report

.